# An Image Detection Model for Aggressive Behavior of Group Sheep

**DOI:** 10.3390/ani13233688

**Published:** 2023-11-28

**Authors:** Yalei Xu, Jing Nie, Honglei Cen, Baoqin Wen, Shuangyin Liu, Jingbin Li, Jianbing Ge, Longhui Yu, Linze Lv

**Affiliations:** 1College of Mechanical and Electrical Engineering, Shihezi University, Shihezi 832003, Chinalvlinze@stu.shzu.edu.cn (L.L.); 2Xinjiang Production and Construction Corps Key Laboratory of Modern Agricultural Machinery, Shihezi 832003, China; 3Key Laboratory of Northwest Agricultural Equipment, Ministry of Agriculture and Rural Affairs, Shihezi 832000, China; 4College of Information Science and Technology, Zhongkai University of Agriculture and Engineering, Guangzhou 510225, China

**Keywords:** behavior detection, deep learning, YOLOv5, GhostNet

## Abstract

**Simple Summary:**

In large-scale meat sheep farming, detecting aggressive behavior in sheep can help to reduce mortality rates and regulate sheep density within the pen. Traditional manual detection methods are time-consuming and laborious, while contact sensor detection methods can be costly and unsuitable for large-scale farming models. In recent years, with advancements in deep learning, a lightweight group sheep for aggressive behavior detection model is proposed. This method can efficiently detect aggressive behavior in group sheep under large-scale farming models.

**Abstract:**

Sheep aggression detection is crucial for maintaining the welfare of a large-scale sheep breeding environment. Currently, animal aggression is predominantly detected using image and video detection methods. However, there is a lack of lightweight network models available for detecting aggressive behavior among groups of sheep. Therefore, this paper proposes a model for image detection of aggression behavior in group sheep. The proposed model utilizes the GhostNet network as its feature extraction network, incorporating the PWConv and Channel Shuffle operations into the GhostConv module. These additional modules improve the exchange of information between different feature maps. An ablation experiment was conducted to compare the detection effectiveness of the two modules in different positions. For increasing the amount of information in feature maps of the GhostBottleneck module, we applied the Inverted-GhostBottleneck module, which introduces inverted residual structure based on GhostBottleneck. The improved GhostNet lightweight feature extraction network achieves 94.7% Precision and 90.7% Recall, and its model size is only 62.7% of YOLOv5. Our improved model surpasses the original model in performance. Furthermore, it addresses the limitation of the video detection model, which was unable to accurately locate aggressive sheep. In real-time, our improved model successfully detects aggressive behavior among group sheep.

## 1. Introduction

In recent years, the demand for meat and milk products has continued to increase, resulting in the expansion of livestock breeding, including sheep, pigs, cattle, and others [1]. Sheep farming has increasingly transitioned towards large-scale production models to reduce costs and increase capacity, raising concerns about the welfare of grazing animals [2]. Sheep aggression is one method to assess the well-being of sheep, as it can cause injury and even death. It is influenced by factors such as sheep density and feed allocation ratios [3,4,5]. Detecting aggression behavior reduces sheep mortality, and whether flock densities and feed ratios are reasonable. However, current sheep behavior detection methods rely primarily on manual observation, which can be inefficient and subjective. Consequently, there is a need to develop sheep aggression recognition models to improve sheep welfare [1,6].

At that time, computer arithmetic was insufficient, and traditional image processing was also one of the main methods for recognizing animal behavior. Simona et al. established a method for detecting the feeding and standing behavior of cows based on the Viola–Jones algorithm [7]. Chen et al. localized aggression events in pigs by connecting area and adhesion index characteristics using rectangular boxes and analyzed the acceleration of the rectangular boxes to determine the level of aggression events [8]. Additionally, Chen et al. used motion shape index in each frame as feature and employed support vector machines to identify aggressive behavior of pigs [9]. However, the aforementioned methods face challenges when it comes to segmenting and extracting animal objects in complex feeding environments, and additional extraction of other motor features may also be required.

With the increase in computational power, deep learning has become widely adopted in the field of agriculture [10,11]. In animal behavior detection, deep learning techniques have eliminated the need for manual extraction of additional features.

Deep learning has been widely used in animal behavior detection, mostly implemented through object detection algorithms. Single-stage object detection algorithms, including the YOLO (You Only Look Once) series [12,13,14,15,16,17] and SSD (Single Shot MultiBox Detector) [18], are commonly applied in animal behavior detection. Additionally, two-stage object detection algorithms like Faster R-CNN [19] are also used. For instance, Zheng utilized Faster R-CNN to recognize the posture of lactating sows [20]. Meanwhile, J. Wang et al. and Joo et al. applied YOLO series of models to identify fighting behaviors in chickens [21,22]. Meanwhile, Thenmozhi et al. used SSD to recognize aggressive and nonaggressive behaviors in sheep [23].

In addition, combining object detection algorithms with other models is a common approach. For example, D. Liu et al. and Shen et al. combined VGG and LSTM (Long Short-Term Memory) to recognize abnormal behaviors in group pigs [1,24]. The LSTM model extracts the temporal information of the feature maps after convolution of the VGG network to binary classify the videos. However, VGG network inference is slow and LSTM model input data are too long. To solve these issues, Xu et al. proposed a YOLOv5-based model to improve the detection speed [25]. In this model, the feature maps information is transformed into animal coordinates, which serve as input to the LSTM model. This time series model incorporates temporal information. None of the aforementioned methods can locate the animal’s position, and they still require human observation to identify abnormal behavior. This limitation hinders subsequent animal detection and tracking. The 3D convolutional video detection model [26], although considering the temporal dimension, is not widely used in animal behavior detection due to its slow detection speed. Alternatively, some researchers combine object detection with lightweight networks to improve network efficiency and detection speed. Yu et al. used YOLOv3 as a detection model for estrus in ewes, with EfficientNet-B0 serving as the backbone of YOLOv3. It also incorporated the SENet attention mechanism to improve detection accuracy [27].

At present, several versions of the YOLO series have been developed, with YOLOv5, YOLOv7, and YOLOv8 being the most widely used ones. All three versions, YOLOv5, YOLOv7, and YOLOv8 are based on the CSPDarkNet53 feature extraction network. In terms of network structure, YOLOv7 proposes the E-ELAN and MPConv modules to enhance the gradient information flow [17]. YOLOv8 replaces the C3 module with the more complex C2f, which provides increased gradient information. Additionally, YOLOv7 introduces the Planned re-parameterized convolution, while YOLOv8 incorporates Segmentation, Pose, and Tracking.

Aiming at large-scale meat sheep farms, aggression behavior is difficult to be detected timely and accurately. A model for detecting aggression in group sheep is presented in this paper Firstly, video data of sheep pens are obtained to form the required dataset. Secondly, GhostNet, MobileNetv3-Large, and ShuffleNetv2 are compared, and we select GhostNet, with the best comprehensive performance, as the feature extraction network of YOLOv5. Additionally, PWConv (Pointwise Convolution) and Shuffle operation are incorporated after the GhostConv (Ghost Module) module to enhance information exchange between feature maps, named PW-GhostConv and CS-GhostConv. Furthermore, a modification to the GhostBottleneck module by replacing its residual structure with an inverted residual structure, named the Inverted-GhostBottleneck. Finally, we conduct an analysis of the advantages and disadvantages of both the image detection model and video detection model.

In summary, the following provides a summary of the contributions made in this article:(1)We replaced the YOLOv5 backbone network with GhostNet to enhance network detection speed and reduce the size of the network model;(2)We propose PW-GhostConv and CS-GhostConv modules to improve the information exchange between feature maps and overcome the issue of information noncirculation after convolution of the GhostConv module.(3)We introduce inverted residual structure in GhostBottleneck to improve the ability of feature extraction;(4)We conducted a comparative analysis of the image detection model and video detection model to evaluate their respective advantages and disadvantages in detecting sheep aggression behavior.

## 2. Materials and Methods

### 2.1. Dataset Collection

The data for the experiments were collected from the Xinao Animal Husbandry Mutton Sheep Breeding Base located in Lanzhou Bay Town, Manas County, Xinjiang Uygur Autonomous Region. Each breeding area consists of an indoor area, an outdoor area, and a shaded area. The layout of surveillance cameras is shown in Figure 1, where the surveillance camera is Hikvision DS-2CD2346FWD-IS Dome Network Camera, resolution 2560 pixels × 1440 pixels, and FPS is 25 f/s.

In this experiment, we developed two models: an image detection model and a video detection model, to study animal aggression behavior in computer vision. However, the research did not include a direct comparative analysis of these two models. Therefore, we established two aggressive behavior detection models to evaluate the advantages and disadvantages of both the image detection model and the video detection model.

#### 2.1.1. Image Detection Model Dataset

We collected 178 videos capturing aggression behaviors among group sheep, which were then decomposed into images using video frame decomposition technique. The videos containing more than 20 sheep in the pen accounted for 20% of the dataset, and 5% of the videos were recorded during nondaytime periods. As a result, we obtained a total of 3218 images of sheep aggression. The images are manually labeled by LabelImg and the label is “aggression”.

#### 2.1.2. Video Detection Model Dataset

The video detection dataset consists of two components: the sheep detection dataset and the group sheep aggression behavior video dataset.

For the sheep detection dataset, we selected the daily behavior of the group sheep. Utilizing the video frame decomposition technique, we obtained 321 images of the daily behavior of the group sheep. These images were manually labeled using the LabelImg and the label is “sheep”.

Additionally, we collected 178 videos of sheep aggression and 129 videos of nonaggressive behavior of sheep. We labelled the videos indicating instances of aggression in group sheep as 1, and those without aggression as 0, as shown in Figure 2.

### 2.2. Data Set Processing

In the image detection model, in order to improve the model’s generalization ability and adaptability to complex environments, we randomly selected half of the data for data augmentation. This includes incorporating ordinary mirroring and rotation scaling operations. Additionally, to enhance the model’s adaptability to various weather conditions, we introduced fogging, rain, and snow effects. The fog generation algorithm is shown in Algorithm 1. Both the rain generation algorithm and the snow generation algorithm required manual generation of random noise. First, the noise was subjected to Affine Transformation and Gaussian blurring to resemble the shape of raindrops and snowflakes. The rotation angle of the noise was adjusted to simulate the angle at which rain and snow occur. Finally, the noise was overlaid onto the original image. The processing results are showed in Figure 3. The dataset was divided into training set, validation set, and test set at a ratio of 6:2:2 [28].
**Algorithm 1:** Fogging algorithm          Require: L: Brightness of the fog          Require: $\theta_{0}$: Fog concentration          Require: img: Image 1: h,w,c←img.shape⊳ Image height, width, number of channels 2: $size\leftarrow \sqrt{\max(h,w)}$⊳ The size of fog 3: for i←0 to h do
 4:        for j←0 to w do
 5:             d←−0.04$\cdot \sqrt{{\left(i-h/2\right)}^{2}+{\left(1-w/2\right)}^{2}}+size$
 6:             $td\leftarrow e^{\left(\theta_{0}\cdot d\right)}$
 7:             $img\left[i\right]\left[j\right]\left[:\right]\leftarrow img\left[i\right]\left[j\right]\left[:\right]\cdot td+L\cdot (1-td)$
 8: end for
 9: return img


To enhance the robustness of the model and increase the size of the sheep detection and group sheep aggression datasets, we applied horizontal, vertical, and diagonal mirroring to both datasets. The sheep detection dataset was randomly split into training, validation, and test sets using ratio of 4:1:1. While examining the aggression videos, we observed that the behavior lasted at least 1 s, and therefore, we divided each video into 1 s subvideos. Overall, we obtained 2126 subvideos of aggressive behavior and 2175 subvideos of nonaggressive behavior. The dataset is divided into training set and test set in 8:2 ratio [24].

### 2.3. Image Detection Model Construction

In this paper, we propose an enhanced model of the YOLOv5 algorithm by replacing the backbone network with the GhostNet network, which draws inspiration from the lightweight architectures of ShuffleNet and MobileNet. To improve the information exchange among different feature maps within the GhostConv module, we introduce two enhanced modules, namely PW-GhostConv and CS-GhostConv. Moreover, we incorporate the inverted residual structure into the GhostBottleneck module to enhance the model’s ability to extract image features.

#### 2.3.1. YOLOv5

Currently, both YOLOv7 and YOLOv8 demonstrate better detection accuracy compared to YOLOv5. However, it is important to consider that increasing the complexity of the network architecture to improve accuracy result in slower network inference speed and increased model parameters. This is particularly unfavorable for the deployment of later algorithm models on mobile devices. Therefore, in this study, we opt for optimizing YOLOv5.

YOLOv5 is based on CSPDarkNet53 as its backbone network. The input image is divided into S × S grids, and each grid randomly assigns three different sizes of anchor boxes. These anchor boxes predict the position and confidence information of the bounding box for each object within the grid. Predicted boxes with low confidence are eliminated by setting a confidence threshold. To avoid the generation of multiple prediction boxes for the same object, the NMS (nonmaximum suppression) algorithm is applied to identify and select the most relevant bounding box for each object.

#### 2.3.2. GhostNet

Huawei Noah’s Ark Laboratory proposed the GhostNet lightweight network [29,30]. In Figure 4a illustrates the output after ordinary convolution, which contains redundant feature maps that still preserve important information from the image. These redundant feature maps can be extracted in subsequent convolutions. However, it can be obtained by a linear transformation operation with lower computational effort and fewer parameters, so the GhostConv module is proposed.

The GhostConv module divides the feature map after convolution into two parts. The first part of the feature maps is obtained by ordinary convolution, while the remaining part is generated by using a linear transformation over the feature map obtained from the previous ordinary convolution. These two types of feature maps are then concatenated as shown in Figure 4b.

In the GhostConv module, Φ denotes the low-cost linear transformation, and $\Phi_{i}$ denotes the feature maps generated by the i-th linear operation. By utilizing this low-cost linear transformation, the module effectively reduces the computational and parameter requirements of the network, while still preserving the feature map information as much as possible.

Suppose the size of the input data is c×h×w, and the GhostConv module generates feature maps of size is $n\times h^{\prime }\times w^{\prime }$. The s denote the ratio of the ordinary convolution to linear transformation, with n/s feature maps generated by ordinary convolution and the remaining n(s − 1)/s feature maps generated by linear transformation. The convolution kernel size of ordinary convolution and linear transformation is k×k and d×d, respectively. When both kernel sizes are the same, the module achieves computational speedup ratio $r_{s}$ and model compression ratio $r_{c}$ as shown in Equations (1) and (2), respectively. In this paper, we take s = 2. Compared to ordinary convolution, GhostConv module can save half of the computation and parameters.
(1)$$r_{s}=\frac{n\cdot h^{\prime }\cdot w^{\prime }\cdot c\cdot k\cdot k}{\frac{n}{s}\cdot h^{\prime }\cdot w^{\prime }\cdot c\cdot k\cdot k+\left(s-1\right)\cdot \frac{n}{s}\cdot h^{\prime }\cdot w^{\prime }\cdot c\cdot d\cdot d}=s$$
(2)$$r_{c}=\frac{n\cdot c\cdot k\cdot k}{\frac{n}{s}\cdot c\cdot k\cdot k+\left(s-1\right)\cdot \frac{n}{s}\cdot d\cdot d}\approx \frac{s\cdot c}{s+c-1}\approx s$$

#### 2.3.3. GhostNet Network Improvements

The YOLOv5 model feature extraction network is based on the improved network of CSPDarkNet53. While GhostNet is another lightweight network, YOLOv5 model can still continue to be optimized in terms of network lightweight. We adopted GhostNet as the feature extraction network for YOLOv5, but with certain modules optimized and improved to enhance the model’s accuracy.

(a)PW-GhostConv and CS-GhostConv

The GhostConv module effectively reduces network computation and parameters compared to Conv while losing some accuracy in the feature maps. This reduction is attributed to lack of information exchange between Identity and the feature maps generated by linear transformation $\Phi_{i}$ in Figure 4b. In the GhostConv module, low-cost linear transformation is the key to reduce computation and parameters. To further enhance the accuracy of the GhostNet network, we propose two improved modules: the PW-GhostConv and the CS-GhostConv modules. These modules focus on optimizing information interchange among feature maps.

To improve information exchange among feature maps following the GhostConv module, we propose adding PWConv and Channel Shuffle operation after the GhostConv module, as shown in Figure 5.

The PWConv uses a 1 × 1 convolution kernel size, which results in reduced computation and parameters. This allows for improving exchange and fusion of information from ordinary convolution and linear transformation of feature maps within the GhostConv module.

The Channel Shuffle operation randomly mixes all feature maps. In the ShuffleNet network, the Channel Shuffle operation solves the problem of nonexchanged feature map information caused by DWConv (Depth Separable Convolution) without bringing any additional computational or parameter cost [31,32]. To enhance the information exchange among the feature maps generated by ordinary convolution and linear transformation, we introduce the Channel Shuffle operation after the GhostConv module.

In the YOLOv5 backbone network, this paper proposes two modules, PW-GhostConv and CS-GhostConv, as replacements for the GhostConv module. However, it is necessary to further explore the performance of these modules and their impact on model detection accuracy, GFLOPs, and memory consumption at different locations during training. This study divides the location of the GhostConv module into two scenarios: single GhostConv module and GhostConv module within the composite GhostBottleneck module. The PW-GhostConv and CS-GhostConv modules are used to replace the GhostConv module in these two locations, respectively. The results of their ablation experiments are presented in Table 1. In the nonbackbone network of YOLOv5, which includes the FPN and PAN multi-branch structure, we select the CS-GhostConv module without computational and parameter to replace the GhostConv module.

Experiments 1–5 all utilize the YOLOv5 model with GhostNet as the feature extraction network, with the GhostConv module, PW-GhostConv module, and CS-GhostConv module as the variable module.

Upon comparing the results of experiments 1, 2, and 3, we observed that only the PW-GhostConv module and the CS-GhostConv module demonstrated improvements in R and mAP when compared to the GhostConv module. Notably, the R metric showed significant improvement, with the PW-GhostConv module having 3% improvement and the CS-GhostConv module having 1.4% improvement. However, there was a slight decrease in P for both modules, and they required more memory during training. It is worth mentioning that the CS-GhostConv module did not introduce any additional computational requirements.

After comparing the results of experiments 2 and 3, it became apparent that the PW-GhostConv module outperformed the CS-GhostConv module. This can be attributed to the superior information fusion and feature extraction capabilities of the PWConv operation in the PW-GhostConv module compared to the Channel Shuffle operation present in the CS-GhostConv module. However, the PWConv operation does require more memory and computation during training than the Channel Shuffle operation.

When comparing experiment 2 with experiment 5 and experiment 3 with experiment 4, it can be observed that replacing the GhostConv module within the GhostBottleneck module has a relatively minor impact on the P of the model. This could be attributed to the fact that the expansion operation following channel compression leads to information loss in the feature maps. Therefore, replacing only one GhostConv module has a minimal overall effect on the results.

From the comparison results between experiment 2 and 5, it can be concluded that it is more beneficial to replace the GhostConv module with a combination of the PW-GhostConv module and the CS-GhostConv module. This approach not only reduces memory consumption during training by 0.55 G, but also decreases computation by 0.7 GFLOPs. 

(b)Inverted-GhostBottleneck

The GhostBottleneck module is an improved lightweight residual structure. It utilizes the GhostConv operation to compress the number of channels to half and then continues with the DWConv operation to extract features while maintaining the same number of channels. The GhostConv operation is then used again to expand the channels and restore the original number, while also adding a shortcut connection between the input and output, as show in Figure 6a. However, this process of channel compression followed by expansion reduces the network parameters but also leads to some loss of feature map information. The GhostConv module and DWConv module significantly decrease the exchange of information among the feature maps.

To reduce feature map information loss in the GhostBottleneck module, we introduce the idea of an inverted residual structure from the MobileNet network [33,34,35]. Unlike the residual structure, the inverted residual structure expands the channels first, extracts features, and finally compresses the channels, preserving more information about the feature map. However, the deep network architecture and parallel structure of YOLOv5 can increase memory costs. Therefore, we set the number of expanded channels to twice the number of original input channels and apply it only to the backbone network extraction. Additionally, we replace the GhostConv module with the cost-free CS-GhostConv module, which enhances information exchange among the feature maps in the DWConv layer, as shown in Figure 6b. By expanding and then compressing the number of channels, the feature map incorporates higher-dimensional information, resulting in increasing amount of feature map information, as shown in Figure 6c. We name the GhostBottleneck module with the inverted residual structure and CS-GhostConv module as the Inverted-GhostBottleneck module, while the GhostBottleneck module with only the CS-GhostConv module is named to as the CS-GhostBottleneck module. The network structure diagram of the group sheep aggressive behavior detection model is shown in Figure 7.

### 2.4. Construction of Video Detection Model

In aggression research, the video detection model is also a mainstream aggression research model, because animals are usually accompanied by movement characteristics when aggression occurs. The coordinate information of all sheep in each frame of the video is fed into the LSTM model through sheep tracking heuristic algorithm, which outputs a single prediction value through the fully connected network [25]. Figure 8 shows the framework diagram of the video detection model.

## 3. Results

### 3.1. Network Training and Evaluation Indexes

The experimental configuration environment for this paper is shown in Table 2. The parameters are set as follows: pre-training weight is eliminated, initial learning rate is 0.01, batch size is 4, Epoch is 300, and the optimizer is SGD.

In practical applications of object detection models, it is crucial to consider not just the model’s detection accuracy, but also its size and speed. Therefore, in this study, we evaluate the model performance using several parameters, including Precision (P), Recall (R), mean average precision (mAP), amount of computation (GFLOPs), model size, F1 score, and detection speed (FPS). Equation (3) shows the calculation of P, R, F1, and mAP.
(3)$$\left\{\begin{array}{l} P=\frac{1}{n}\sum_{i=1}^{n}\frac{TP_{i}}{FP_{i}+{TP}_{i}}\\ R=\frac{1}{n}\sum_{i=1}^{n}\frac{TP_{i}}{FN_{i}+{TP}_{i}}\\ \mathrm{mAP}=\frac{\sum_{i=1}^{n}\int_{0}^{1}P_{i}\left(R_{i}\right)dR_{i}}{n} \end{array}\right.$$
where n is the number of detected classes, i is the detected class, $TP_{i}$ is the true positive, which represents the number of samples correctly identified by i class. $FP_{i}$ is the false positive, which represents the number of samples incorrectly identified other classes as i class. In addition, $FN_{i}$ is the false negative, which represents the number of samples that i class is not correctly identified.

### 3.2. Evaluation Indexes Analysis

To explore the variation in each index with training time, the variation curves of P, R, mAP, and Loss value of the improved model in this paper with training time are plotted in Figure 9.

Figure 9a shows the curves of the loss function during the training process of the improved model. It can be observed that the loss function converges to 0.02 after 250 epochs. From Figure 9b, it can be shown that P, R, and mAP converge at 0.95, 0.91, and 0.95, respectively, after 250 Epochs. Because the features of the sheep with aggression in the case of very dense flocks are easily obscured by the nonaggressive sheep, which leads to false detection, resulting in R lower than P.

Figure 9c shows the comparison of R in the training process of different models. It can be observed that the YOLOv5 model has a higher initial growth rate of R and is relatively heavier in terms of weight. However, after 230 epochs, there is no significant difference in the performance of the YOLOv5 model compared to improve the model in the paper.

### 3.3. Experimental Results Analysis

In recent years, researchers have proposed various lightweight models, such as MobileNetv3, ShuffleNetv2, and GhostNet. In our research, we replaced the feature extraction network of YOLOv5 with these models and compared their performance with the improved model proposed in this paper, as well as with the YOLOv5 and SSD models. The experimental results are shown in Table 3.

The experimental results indicate that GhostNet outperforms ShuffleNetv2 and MobileNetv3-Large in terms of overall performance when selecting a lightweight feature extraction network for the YOLOv5 model. Compared to ShuffleNetv2, GhostNet demonstrates superior performance in P, output weight, and detection speed. Although MobileNetv3-Large has the highest P, its feature extraction network comprises multiple inverted residual structures, which require significant channel compression (more than 2–6 times). This compression significantly reduces the model’s detection speed, increases training time, and consumes more memory. Therefore, GhostNet was selected as the feature extraction network to enhance the YOLOv5 model.

The improved model proposed in this study demonstrates excellent performance in terms of detection accuracy, output weight, and detection speed. When compared to the YOLOv5 model, R and mAP are improved by 2% and 0.7%, P decreases slightly, faster detection speed, and the model size is 62.7% of YOLOv5. Compared with the SSD model, P is reduced by 1.3%, but R and mAP are improved by 3.9% and 1%, and the model is less than one-tenth that of SSD.

Table 4 shows the experimental results of the video detection model, showing that it achieves P of 93.33% and R of 91.74% on the test set. Notably, the video detection model exhibits better performance than the image detection model in terms of R.

## 4. Discussion

### 4.1. Model Comparison Analysis

#### 4.1.1. Image Detection Model Comparison

To illustrate the advantages of this paper’s model over other models, we evaluated the model’s detection performance in three different scenarios: multiple aggression events and aggression event under dim light separately evaluated. The results of these tests are showed in Figure 10, Figure 11 and Figure 12, where the red box is the aggressive behavior detection box, the upper left corner of the red box is the name of the behavior class, and the upper right corner is the confidence of the behavior.

In single aggression events, the models have good detection results. However, when multiple aggression events occur and in dim light conditions, the model shows misdetection and missed detection.

Sheep are easily susceptible to multiple aggressions when the population density of sheep is too high. The chaotic environment further increases the difficulty of detection. As show Figure 9, two aggression events with three sheep were recorded. From Figure 11, we can be known that SSD, YOLOv5-GhostNet, and the model proposed in this paper have all successfully detected the two aggression events. However, YOLOv5 did not generalize as well as the lightweight model in multiple aggressions and did not successfully detect them. YOLOv5-ShuffleNetv2 and YOLOv5-MobileNetv3-Large show missed and false detections.

Figure 12 illustrates the detection effects of different models in dim light conditions. YOLOv5-ShuffleNetv2 shows missed detection, while YOLOv5-MobileNetv3-Large shows false detection again. YOLOv5, SSD, YOLOv5-GhostNet, and the model proposed in this paper are successfully detected.

#### 4.1.2. Video Detection Model Comparison

The video detection model to extract the motion features of the animal has a faster detection speed compared to 3D convolution. However, its performance relies heavily on the accuracy of YOLOv5 for detection and the variability of the motion features. In Figure 13, the left side illustrates aggression behavior of group of sheep occurring at the edge of the monitoring field of view, while the right side showcases courtship behavior among the sheep. Figure 13a shows the image detection model’s results, and Figure 13b shows the video detection model, and the image detection model successfully performs the recognition. In the video detection model, the detection frame position of the sheep at the edge position of the field of view has a large error with the actual position and is not successfully recognized. The sheep’s courtship behavior is accompanied by chasing motion characteristics, which is similar to the aggression behavior motion characteristics, and misdetection occurs. The most important problem of the video detection model in this paper is that it cannot locate the aggressive sheep.

In Figure 13, the left side show group of sheep engaging in aggressive behavior at the edge of the monitoring field of view, while the right side showcases courtship behavior among the sheep. Figure 13a shows the results of the image detection model, whereas Figure 13b showcases the outcomes of the video detection model. The image detection model successfully recognizes the aggressive behavior. However, the video detection model presents a significant error in detecting the sheep’s position at the edge of the field of view, resulting in unsuccessful recognition. Additionally, due to the similarities in chasing motion characteristics between courtship behavior and aggression behavior, the video detection model often to misjudge such instances. The primary issue with the video detection model proposed in this research is its inability to accurately locate aggressive sheep.

### 4.2. Network Visualization

To verify the actual effectiveness of the PW-GhostConv, CS-GhostConv, and Inverted-GhostBottleneck modules proposed by the model in the network, we generated heat maps using Grad-CAM and compared them with the heat maps of YOLOV5-GhostNet and YOLOv5. The heat map was uniformly selected from the Detective layer heat map of the shallowest layer of the network, as shown in Figure 14. A brighter color indicates that the model is more sensitive at that particular location.

As observed in Figure 14, the heat map of the proposed improved model in this paper focuses on the aggressive sheep, particularly highlighting the head impact characteristics of the aggressive sheep.

## 5. Conclusions

We compared lightweight GhostNet, ShuffleNetv2 and MobileNetv3-Large networks, and selected GhostNet as the feature extraction network for YOLOv5 by the comparison results. The GhostConv module in the GhostNet network reduces the number of model parameters and computation by linear transform and ordinary convolution. However, it also blocks the information fusion of the feature maps after these two parts of convolution. We added computationally less PWConv and computationally free Channel Shuffle operation after GhostConv to improve the information flow of the feature map, and place PW-GhostConv and CS-GhostConv in reasonable positions in the YOLOv5 backbone network through ablation experiments. Additionally, we replaced the residual structure in GhostNet network with inverted residual structure. With these improvements, our proposed model not only models smaller than YOLOv5, but also has better detection accuracy than the YOLOv5 model. The improved model P, R, and mAP values of 94.7%, 90.7%, and 95.5% respectively. Real-time detection of group sheep aggression is successfully achieved.

Subsequent work can be further explored in extracting image information while retaining faster detection speeds to locate aggressive behavior in sheep.

## Figures and Tables

**Figure 1 animals-13-03688-f001:**
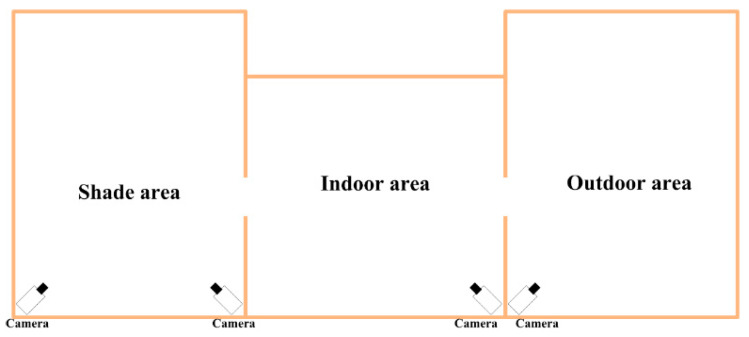
Surveillance Camera Layout Diagram.

**Figure 2 animals-13-03688-f002:**
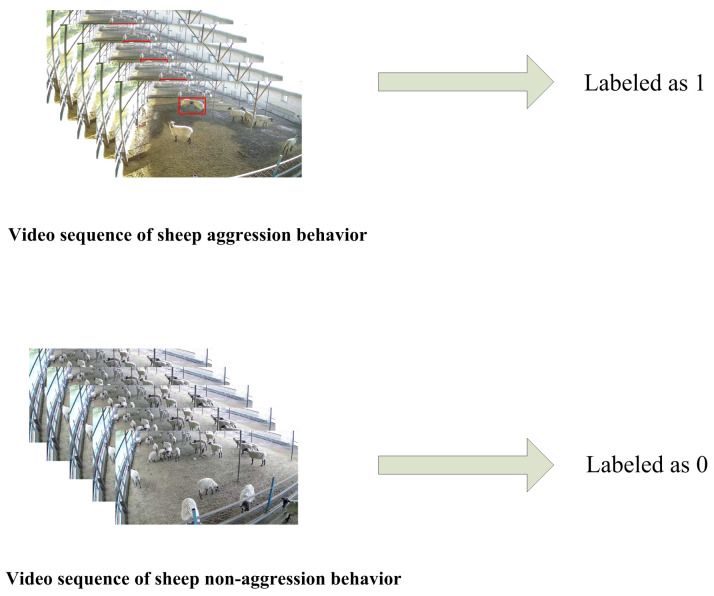
Video data labeling. Red boxes are sheep aggression events.

**Figure 3 animals-13-03688-f003:**
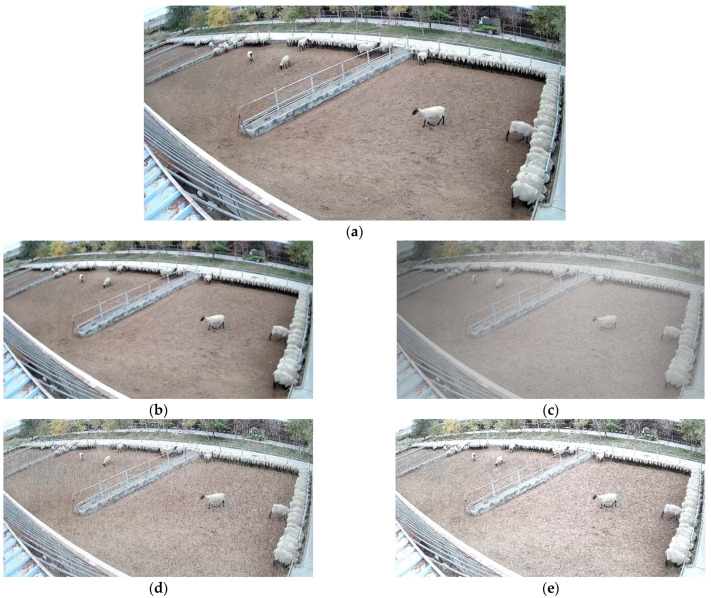
Data augmentation: (**a**) Original; (**b**) Gaussian blurring; (**c**) Fog; (**d**) Rain; (**e**) Snow.

**Figure 4 animals-13-03688-f004:**
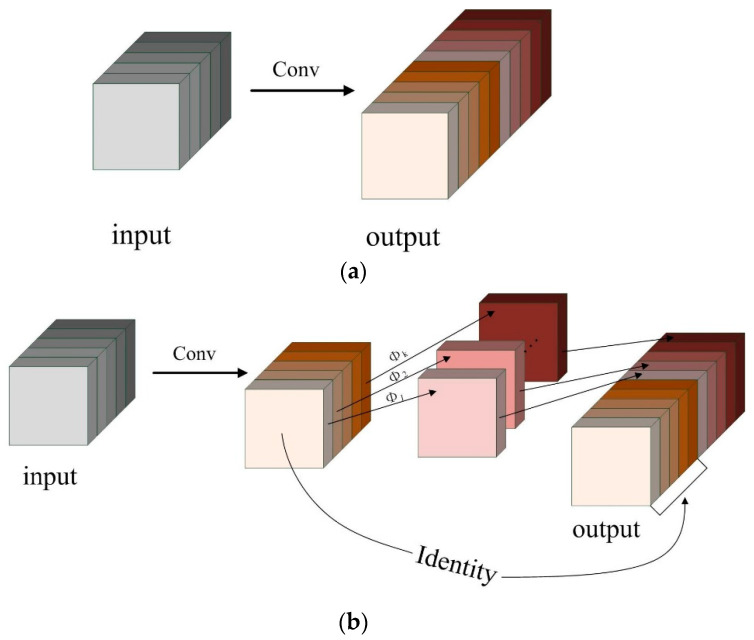
Comparison of different convolutions: (**a**) Conv (Convolution); (**b**) GhostConv.

**Figure 5 animals-13-03688-f005:**
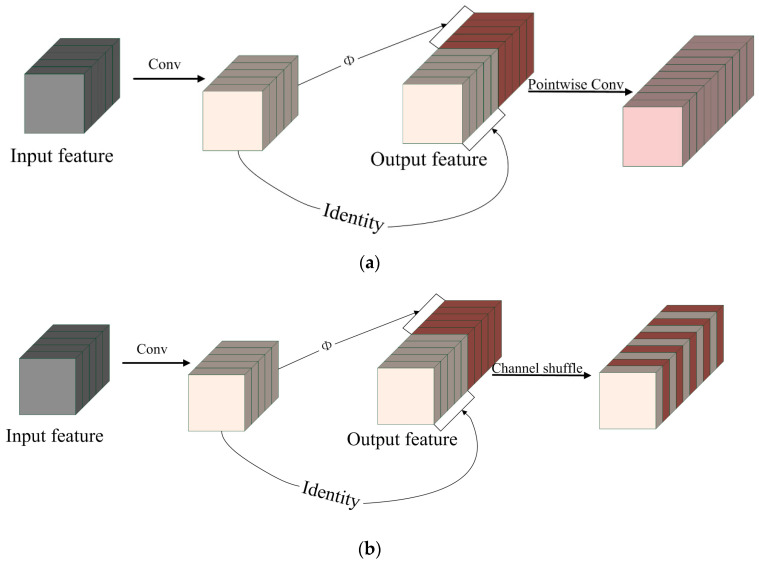
PW-GhostConv and CS-GhostConv: (**a**) PW-GhostConv; (**b**) CS-GhostConv.

**Figure 6 animals-13-03688-f006:**
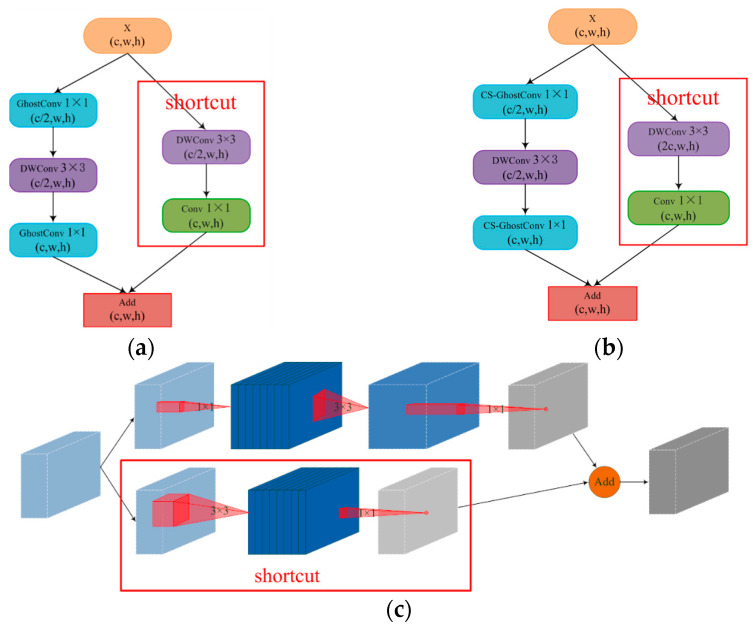
GhostBottleneck and Inverted-GhostBottleneck struct: (**a**) GhostBottleneck; (**b**) Inverted-GhostBottleneck; (**c**) Inverted-GhostBottleneck.

**Figure 7 animals-13-03688-f007:**
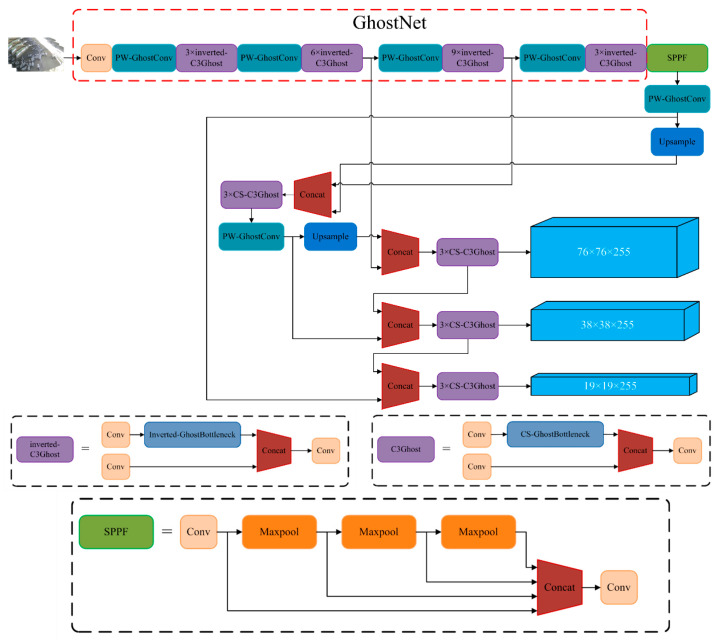
Improved YOLOv5 network structure diagram.

**Figure 8 animals-13-03688-f008:**
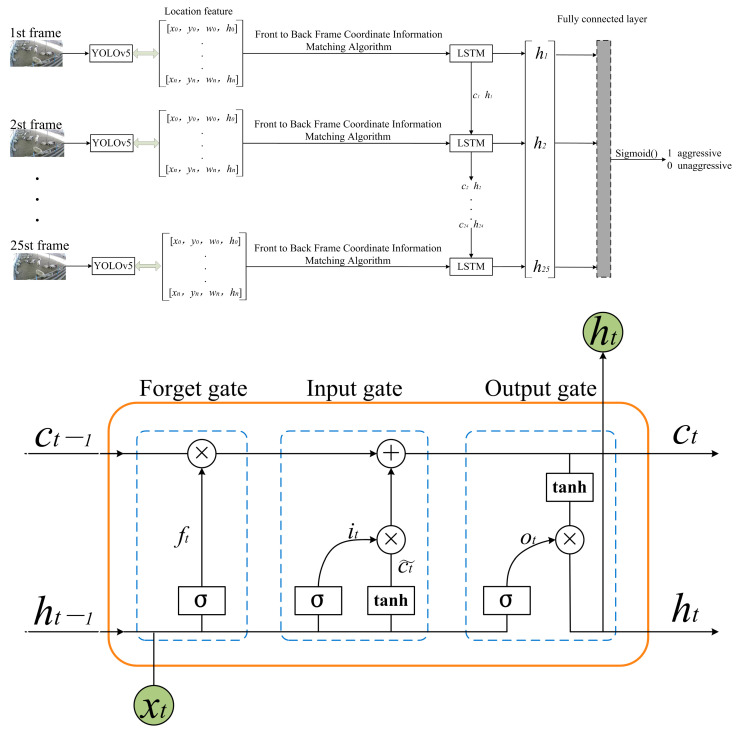
YOLOv5-LSTM model structure. $f_{t}$ is forgetting gate, $i_{t}$ is input gate, $O_{t}$ is output gate, and σ is activation function Sigmod().

**Figure 9 animals-13-03688-f009:**
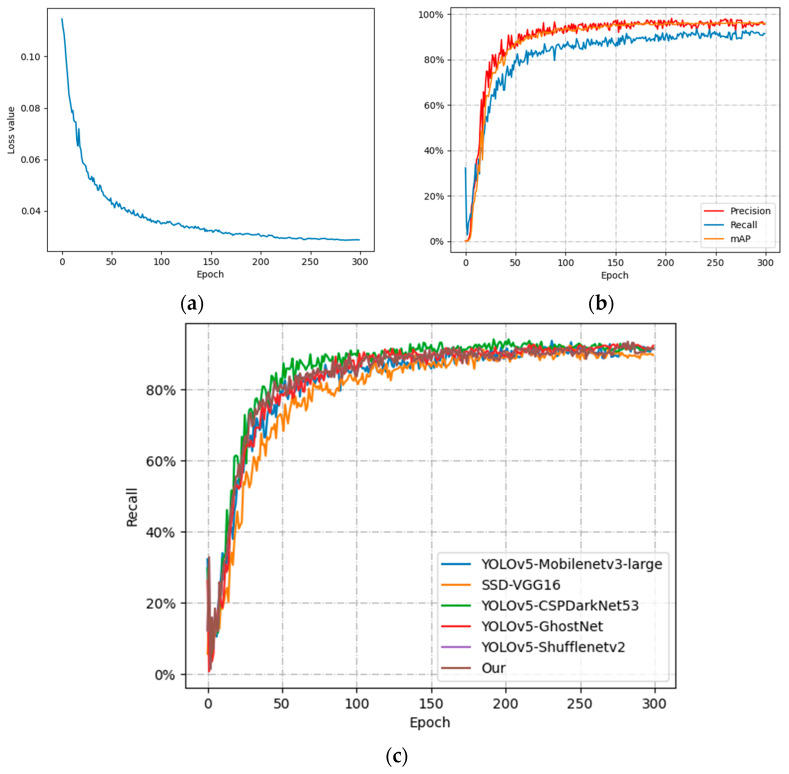
Variation curves of each index under different backbone networks: (**a**) Loss-Epoch; (**b**) P, R, mAP-Epoch; (**c**) R-Epoch for different models.

**Figure 10 animals-13-03688-f010:**
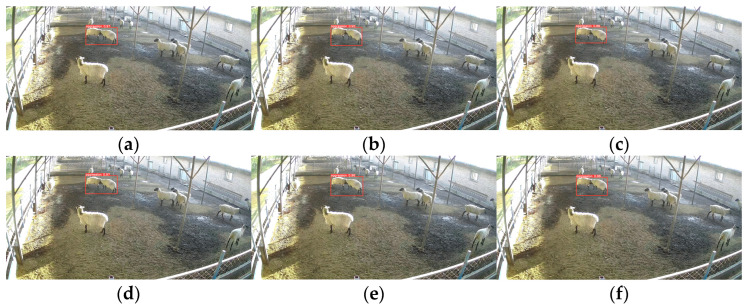
Comparison chart of single aggression event models for sheep: (**a**) YOLOv5; (**b**) SSD; (**c**) YOLOv5-ShuffleNetv2; (**d**) YOLOv5-MobileNetv3-Large; (**e**) YOLOv5-GhostNet; (**f**) Our Model.

**Figure 11 animals-13-03688-f011:**
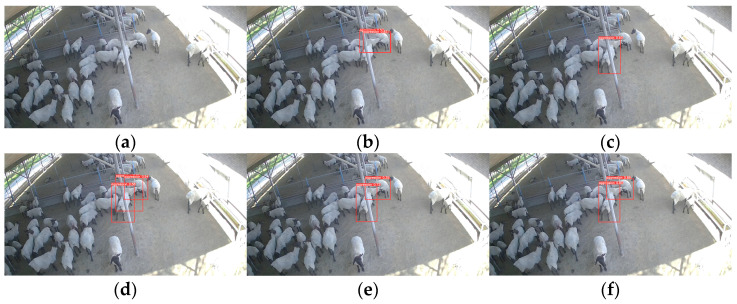
Comparison chart of multiple aggression event models for sheep: (**a**) YOLOv5; (**b**) SSD; (**c**) YOLOv5-ShuffleNetv2; (**d**) YOLOv5-MobileNetv3-Large; (**e**) YOLOv5-GhostNet; (**f**) Our Model.

**Figure 12 animals-13-03688-f012:**
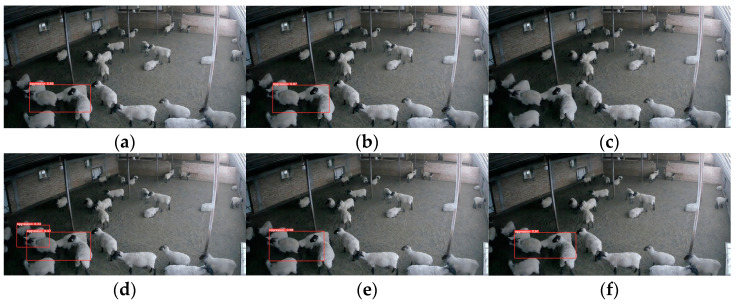
Comparative chart of aggression event model for sheep in dim light: (**a**) YOLOv5; (**b**) SSD; (**c**) YOLOv5-ShuffleNetv2; (**d**) YOLOv5-MobileNetv3-Large; (**e**) YOLOv5-GhostNet; (**f**) Our Model.

**Figure 13 animals-13-03688-f013:**
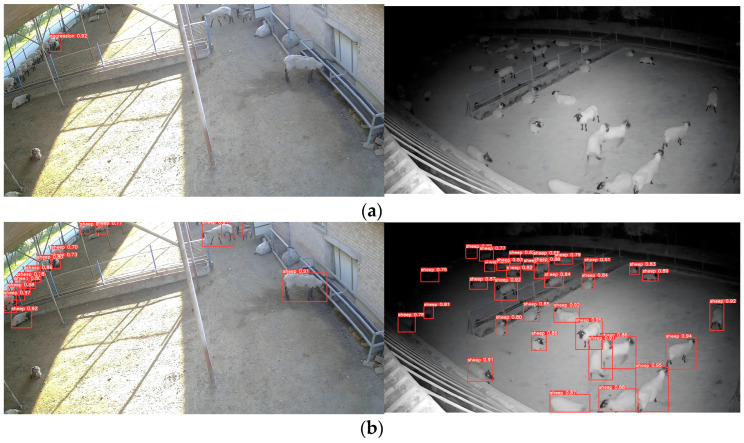
Comparison of Image Detection Model and Video Detection Model: (**a**) Image Detection Model; (**b**) Video Detection Model.

**Figure 14 animals-13-03688-f014:**
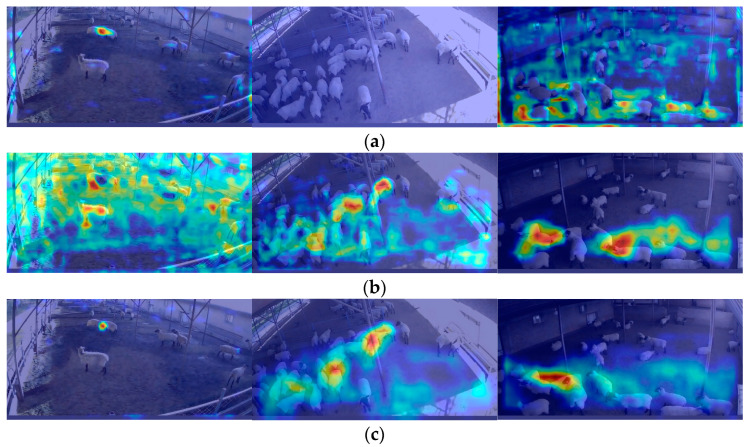
Heat map comparison: (**a**) YOLOv5s; (**b**) YOLOv5-GhostNet; (**c**) Our Model.

**Table 1 animals-13-03688-t001:** Ablation experiment.

Structure	Memory (G)	P (%)	R (%)	mAP (%)	GFLOPs
GhostConv + GhostBottleneck(GhostConv)	0.805	95.4	85.1	93.7	8.2
PW-GhostConv + GhostBottleneck(PW-GhostConv)	1.07	95.1	88.1	94.6	10.0
CS- GhostConv + GhostBottleneck(CS-GhostConv)	0.837	95.2	86.5	94.1	8.2
CS-GhostConv + GhostBottleneck(PW-GhostConv)	0.969	95.0	86.9	94.3	9.0
PW-GhostConv + GhostBottleneck(CS-GhostConv)	0.952	95.1	88.0	94.4	9.3

**Table 2 animals-13-03688-t002:** Experimental configuration environment.

Configuration	Parameter
CPU	AMD Ryzen 7 5800H
GPU	NVIDIA GeForce RTX 3050
Operating system	Windows 11
Development environment	Pycharm 2021

**Table 3 animals-13-03688-t003:** Comparisons of different algorithms.

Model	Backbone	P (%)	R (%)	mAP (%)	Weight (MB)	FPS (f/s)
YOLOv5	CSPDarkNet53	95.4	88.7	94.8	13.7	129.9
SSD	Vgg16	96.0	86.8	94.5	92.6	53.2
YOLOv5	ShuffleNetv2	96.2	84.1	92.6	7.6	153.9
YOLOv5	MobileNetv3-Large	97.0	85.1	94.3	8.8	90.9
YOLOv5	GhostNet	95.4	85.1	93.7	7.4	161.3
Ours	Improvement-GhostNet	94.7	90.7	95.5	8.6	147.1

**Table 4 animals-13-03688-t004:** Test results of the video detection model.

	Prediction	Positive	Negative
Reference			
Positive		395	35
Negative		28	427

## Data Availability

The data presented in this study are available on request from the corresponding author. The data are not publicly available due to being part of an ongoing study.

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
