# Peer review of "An Image Detection Model for Aggressive Behavior of Group Sheep"

_animals, 2023, doi:10.3390/ani13233688_

Round 1

Reviewer 1 Report

Comments and Suggestions for Authors

The authors have conducted an interesting research project on a lightweight group sheep aggression behavior recognition modeling approach. However, before considering it for publication, there are several issues that need to be addressed in the paper.

 Title:
Please mention your method of recognition within the title.

Introduction

There are several research papers that apply YOLOv7 and YOLOv8 for classification and sorting applications. It would be beneficial to address some of these papers in this section.

In the last paragraph please explain about your novelties.

Material and Method.
Why have you selected YOLOv5 to optimize?

It is recommended that you compare your results with more recent and advanced models such as YOLOR, YOLOv7, and YOLOv8. Additionally, it would be beneficial to compare not only classification accuracy and precision but also the time of detection.

Furthermore, what your method does while the ship is out of the video sequence? 

Results
The discussion section of your paper requires further development. Please provide a more detailed explanation of the reasons why your proposed method achieves better results than other methods. 

 Why your method outperformed compare to other algorithm?   

Reviewer 2 Report

Comments and Suggestions for Authors

General comment

This paper presents an interesting deep learning-based method for detecting sheep aggressive behavior. It has certain novelty and contributes some values to the community. Although the language and structure must be improved, I enjoyed reading this paper. Please find below my specific suggestions and comments.

Abstract
Line 23: As this is not the first investigation of computer vision on the detection of sheep aggressive behaviors, it is better to state here the research gap that you wanted to fill (relative to previous works) before you start presenting your work.
Line 26: ", respectively. The PW-GhostConv and CS-GhostConv modules improved the information exchange among different feature maps, and ablation experiment was conducted…."
Line 30: I suggest changing “proposed” to “applied” since it is an existing module instead of a newly proposed one in this work.
Line 32: To the end of the abstract, here you miss some more words to show the relevance and perspective of your work.

Introduction
There is a bunch of “objective detection” which I believe should be “object detection”, please modify.

Line 52: I doubt the reference style used here. Normally it should be something like “Alvarenga et al. [9]”, please confirm with the instructions for authors.

Line 55: “Shen et al. identified….”

Line 89: What is Xu et al. cited for? Unclear writing here. Either remove “Xu et al.” or modify as “Xu et al. proposed a YOLOv5-based model to improve the detection speed.”

Line 99: The writing of the last sentence here is really bad, please modify.

Materials and methods
Line 130: Three datasets? Although you explained latter these three datasets, it is really confusing to put it here, I suggest removing this sentence. Also please add here the motivation of having these two models.
Line 135: I was thinking it may be better if you could add a description of the sample size of each kind of aggressive behavior. It may help explain latter in which cases the model has the worst result.
Line 154: What do the authors mean by “fogging algorithm takes the image as shown in Table 1.” Are you citing Table 1 or Figure 2?

Line 250: I suggest the authors to clearly explain their design for the comparison and ablation studies to be conducted. It should be in M&M instead of Results.

Results
Line 290: This part should definitely be put into M&M, please rearrange.

Discussion
It is not normal to introduce new results (especially like Figure 8) in this section, please rearrange.

Figure8: it is confusing since GhostNet, ShuffleNetV2, and MobileNetV3 were the backbones to compare. Please try to make this caption more precise.

Conclusions
Please remove phrases like “To implement a lightweight detection model of group sheep aggression” and “To improve the information flow of different feature maps of GhostConv module”, just show the readers the most significant results and their implications of this work.

It would be better to provide a public link or depository to the datasets used in this study, as the author stated that one of the contributions of this work is the dataset.

Comments on the Quality of English Language

The manuscript contains a bunch of grammatical errors and unclear expressions, thus requiring thorough polish.

Reviewer 3 Report

Comments and Suggestions for Authors

1. In the 2.1 Dataset collection, it is defined that "aggregation occurred in the group of sheet as 1, and that in which no aggregation occurred in the group of sheet as 0." It is recommended to add images here to enhance readers' visibility.

2. In 2.3.1 YOLOv5, the author uses YOLOv5s, but YOLOv5n seems to be more suitable for deployment and lightweight. Did the author make a comparison?

3. In 3.1 Network training and evaluation indexes, the author chose SGD as the optimizer and adam when the data volume is not large. Please explain why they chose adam?

4. Please increase the clarity of Figures 9, 10, and 11 in order for readers to better obtain information from them.

5. In Table 4 Comparisons of different algorithms, adding units from the table to the header is more appropriate.

6. The units of P, R, and mAP in Figure 8 should be consistent with those in the table.

7. Suggest adding discussion in the paper 。

Comments on the Quality of English Language

Minor editing of English language required

Reviewer 4 Report

Comments and Suggestions for Authors

The author proposed a lightweight group aggression detection model based on YOLOv5, which can satisfy the detection of group sheep aggression behavior under 20 

the scale farming mode. But there are still a few points that are not clear and I hope that they can be further explained.

  1. Page 4, line 151. After data enhancement, how many images are there in total in the dataset? 
  2. Page 8, line 271. It would be better to give an explanation of the parameter s.
  3. Page 11, line 284. The LSTM model used in this paper should be explained. 
  4. Part 3. In Result part, the ablation experiments should include the Inverted residual structure you introduced above. 
  5. Part 3, Results. It seems that you only discussed the image detection model, how about the results of the video detection model?
  6. In this paper you mainly talked about the image detection model, what is the significance and usefulness of the video monitoring model you mentioned?
  7. According to the context, the annotation of d in Figure 2 should be 'rain' instead of 'train'.
Comments on the Quality of English Language

none

Round 2

Reviewer 2 Report

Comments and Suggestions for Authors

Thank the authors for providing point-to-point response.

I saw your reply (duplicated below) but found nothing actually changed in the document. Please clarify.

Line 290: This part should definitely be put into M&M, please rearrange. Answer: Thanks to anonymous reviewers for their valuable comments. We finish the modifications.

Comments on the Quality of English Language

Major language issues remained.
